# Structural Design, Synthesis and Antioxidant, Antileishmania, Anti-Inflammatory and Anticancer Activities of a Novel Quercetin Acetylated Derivative

**DOI:** 10.3390/molecules26226923

**Published:** 2021-11-17

**Authors:** Saul Vislei Simões da Silva, Orlando Maia Barboza, Jéssica Teles Souza, Érica Novaes Soares, Cleonice Creusa dos Santos, Luciano Vasconcellos Pacheco, Ivanilson Pimenta Santos, Tatiana Barbosa dos Santos Magalhães, Milena Botelho Pereira Soares, Elisalva Teixeira Guimarães, Cássio Santana Meira, Silvia Lima Costa, Victor Diógenes Amaral da Silva, Lourenço Luís Botelho de Santana, Aníbal de Freitas Santos Júnior

**Affiliations:** 1Department of Life Sciences, State University of Bahia (UNEB), Salvador 41150-000, BA, Brazil; saulvislei@gmail.com (S.V.S.d.S.); orlandmb0@gmail.com (O.M.B.); lucianofcd@hotmail.com (L.V.P.); tatibasa@hotmail.com (T.B.d.S.M.); etguimaraes@uneb.br (E.T.G.); cassio.meira@fieb.org.br (C.S.M.); lourencoluisbotelho@gmail.com (L.L.B.d.S.); 2Laboratory of Neurochemistry and Cell Biology, Department of Biochemistry and Biophysics, Federal University of Bahia, Salvador 40231-300, BA, Brazil; telles.jessica@hotmail.com (J.T.S.); ericanovaessoares@gmail.com (É.N.S.); cleonicemev@gmail.com (C.C.d.S.); costasl@ufba.br (S.L.C.); vdsilva@ufba.br (V.D.A.d.S.); 3Gonçalo Moniz Institute, FIOCRUZ, Salvador 40296-710, BA, Brazil; eagle_nito@hotmail.com (I.P.S.); milena@bahia.fiocruz.br (M.B.P.S.); 4SENAI Institute of Innovation in Health Advanced Systems (CIMATEC ISI SAS), University Center SENAI/CIMATEC, Salvador 41650-010, BA, Brazil

**Keywords:** quercetin, synthesis, quercetin pentaacetate, antioxidant, antileishmania, anti-inflammatory, cytotoxicity activity

## Abstract

Quercetin (Q) is a bioflavonoid with biological potential; however, poor solubility in water, extensive enzymatic metabolism and a reduced bioavailability limit its biopharmacological use. The aim of this study was to perform structural modification in Q by acetylation, thus, obtaining the quercetin pentaacetate (Q5) analogue, in order to investigate the biological potentials (antioxidant, antileishmania, anti-inflammatory and cytotoxicity activities) in cell cultures. Q5 was characterized by FTIR, ^1^H and ^13^C NMR spectra. The antioxidant potential was evaluated against the radical ABTS^•+^. The anti-inflammatory potential was evaluated by measuring the pro-inflammatory cytokine tumor necrosis factor (TNF) and the production of nitric oxide (NO) in peritoneal macrophages from BALB/c mice. Cytotoxicity tests were performed using the AlamarBlue method in cancer cells HepG2 (human hepatocarcinoma), HL-60 (promyelocytic leukemia) and MCR-5 (healthy human lung fibroblasts) as well as the MTT method for C6 cell cultures (rat glioma). Q and Q5 showed antioxidant activity of 29% and 18%, respectively, which is justified by the replacement of hydroxyls by acetyl groups. Q and Q5 showed concentration-dependent reductions in NO and TNF production (*p* < 0.05); Q and Q5 showed higher activity at concentrations > 40µM when compared to dexamethasone (20 µM). For the HL-60 lineage, Q5 demonstrated selectivity, inducing death in cancer cells, when compared to the healthy cell line MRC-5 (IC_50_ > 80 µM). Finally, the cytotoxic superiority of Q5 was verified (IC_50_ = 11 µM), which, at 50 µM for 24 h, induced changes in the morphology of C6 glioma cells characterized by a round body shape (not yet reported in the literature). The analogue Q5 had potential biological effects and may be promising for further investigations against other cell cultures, particularly neural ones.

## 1. Introduction

Quercetin is a bioflavonoid with a proven impact on health and well-documented biochemical activities. This compound is considered one of the most potent antioxidants among polyphenols [1,2,3]. Due to its properties, quercetin has been tested for various therapeutic applications, such as antioxidant, antiparasitic, anti-inflammatory and anticancer activities [4,5]. In parasitic diseases, flavonoids are a high interest group. This is due to their low toxicity in hosts and several mechanisms by which they can modulate pathologically altered processes during infections [6]. Flavonoids have multiple targets for treating leishmaniasis and include targets, such as arginase, ribonucleotide reductase and topoisomerase II [7,8].

Quercetin has multiple anticancer activities in several types of solid tumors. Furthermore, it has been proven to have activity in HL-60 cells (originating from acute myeloid leukemia (AML)) to significantly reduce tumor growth by reducing intratumoral oxidative stress, activation of the Extracellular Regulated Kinase (ERK) signal and subsequent apoptosis [9]. Quercetin improved the pro-inflammatory response by reducing IL-6 and TNF expression with positive anti-inflammatory activity in human THP1 macrophage populations [10].

Among the different types of cancer, glioma is one of the most aggressive and with poor prognosis. Unfortunately, main therapies (surgical removal, radiation therapy and chemotherapy) are not efficient. The average of overall survival for patients remains at approximately 14 months [11]. Nevertheless, new compounds, such as temozolomide [12], retinoids [13] and flavonoids [14] have shown anti-glioma activity. The antitumoral effects of quercetin have been described against cells from glioblastoma, which is the most aggressive of the gliomas [15].

Abnormal immune responses are involved in the initiation and development of a large number of diseases, including autoimmune diseases, allergies, cancer and neurodegenerative diseases. Flavonoids have potential immunomodulatory activity and are studied as alternatives for clinical use. Quercetin can exert significant immunomodulatory effects on the cellular production of cytokines derived from the Th-1 and Th-2 profile and lymphocytic proliferation, regulating cellular immunity [16]. Furthermore, quercetin can differentially modulate the expression of interleukin genes in peripheral blood mononuclear cells (PBMC), favoring the Th-1 profile, which promotes cellular immunity by interferon gamma (IFN-γ), the reduction of the Th-2 profile involved in humoral immunity and macrophage activation by IL-4 [17].

The l-ow water solubility, extensive metabolism and enzymatic degradation limit bioavailability and reduce the biopharmacological use of quercetin as a therapeutic agent [13]. An interesting approach to overcome the low bioavailability of polyphenols, in order to test and explore their in vivo activity, is the chemical modification of the natural compound, increasing solubility and slowing down metabolism. Thus, many synthetic routes are being studied, such as acetylation, addition of amines and bromination, modification to oximes and complexation with hydrazines [18,19,20].

In this study, we perform a structural modification in quercetin, aiming to obtain the analogue quercetin pentaacetate (Q5) for evaluation of the antioxidant potential, anti-inflammatory activity and inhibition of the growth of cancer cells, hepatocarcinoma (HepG2), promyelocytic leukemia (HL-60) and rat glioma (C6) cells.

## 2. Results and Discussion

### 2.1. Synthesis and Characterization

For the synthesis of the analogue quercetin (3,3′,4′,5,7-pentaacetate), the total acetylation approach was used, which was adapted to the experimental conditions found in the literature [21]. In this way, acetic anhydride is preserved as an acylating agent and pyridine as a catalyst.

The product obtained (Q5), a yellow solid, was characterized featuring melting points in the range of 178–186 °C, compatible with the literature data [18]. Additionally, the comparative analysis of the FTIR between quercetin and the product demonstrated the absence of absorption bands in 3400 cm^−1^ characteristic of quercetin hydroxyls and by the appearance of bands around 1600–1650 cm^−1^ indicative of carbonyl ester, demonstrating the replacement of hydroxyl groups by acetyl groups: IR (KBr) ν (cm^−1^): 1761, 1652, 1615, 1505, 1442, 1377, 1208, 1176, 1121, 1085, 1012, 892, 837, 691 and 600 (Figure 1). The spectra were comparable with those described in the literature [21].

Nuclear magnetic resonance (NMR) analyses of the quercetin (Q) and product obtained (Q5) showed total acetylation, with the disappearance of singlets characteristic of hydroxyls above 9 ppm in the ^1^H NMR spectrum (Appendix A) and the appearance of large and characteristic signals of methyl in the aliphatic regions of the spectrum, between 2 and 3 ppm. For the ^13^C NMR spectrum (Appendix A), the product obtained showed an increase in the representative signals of the ester carbonyls close to 170 ppm and the arising of signals between 20 and 21 ppm corresponding to the chemical shifts of the five aliphatic carbons of methyls, verified through the integral of the signals. The spectra were comparable with those described in the literature [22].

Biasutto et al. [23] were pioneers in the investigation of precursors based on esters, in order to increase the bioavailability of quercetin. Based on their studies, Mattarei et al. [21] promoted the total acetylation of quercetin and obtained the Q5 derivative in high yield (79–97%). More recently, Mohajeri et al. [24] obtained a yield of 85% in obtaining analogue Q5, under heating (180 °C for 6 h). In the present study, the synthesis of Q5 was efficient, and we obtained one compound duly characterized according to the data described in the literature.

### 2.2. Antioxidant ABTS^•+^ Radical Activity

The comparative analysis of the antioxidant activity of Q and Q5 showed that quercetin (Q) was more active in scavenging the ABTS^•+^ radical than Q5 (29% and 18%, respectively), with IC_50_ values (in µM) of 188.850 ± 0.003 and 379.560 ± 0.004 for Q and Q5, respectively.

The antioxidant activity of flavonoids is directly related to their molecular structure. There are two mechanisms through which phenolic compounds can exert their antioxidant functions: hydrogen atom transfer and electron donation [25]. Quercetin’s superior antioxidant activity can be attributed to the significant contribution of hydroxyls and their hydrogens, which are replaced by acetyls in Q5 analogue, thus, decreasing the capacity for hydrogen donation or free radical scavenging by synthetic molecules.

No data were found in the scientific literature reporting the antioxidant activity of the Q5 compound. Oh et al. [26] evaluated quercetin ester preparations and their antioxidant activities, showing that quercetin had the highest radical scavenging activity among the tested samples. Therefore, the high contribution of the hydroxyl groups in the antioxidant activity of quercetin are essential [27].

### 2.3. Antileishmania Activity

In order to evaluate the activity of the compound (Q and Q5) against the promastigote forms of the two Leishmania species, the 50% inhibitory concentration (IC50) was calculated from the cell viability assay in axenic culture. For Leishmania braziliensis, Q and Q5 had IC_50_ values > 100 μM when compared to amphotericin B (IC501.1 μM ± 0.1). In addition, the effects of Q5 on the promastigote forms of L. amazonenses were evaluated, and this compound exhibited a lower IC_50_ value (75.1 ± 4.7 μM) for this species compared to L. braziliensis.

Tasdemir et al. [8] compared the antitrypanosome and antileishmanial activities of flavonoids and their analogues (derived from quercetin) and found them to be potent and effective antiprotozoal agents against amastigote forms of L. donovani (IC_50_ = 1.0 µg mL^−^^1^). For promastigote forms of L. amazonensis, Fonseca and Silva et al. [27] found IC_50_ = 31.4 µM in cultures treated with quercetin within 48 h. Furthermore, they reported complete cell growth arrest with 96 µM quercetin in 96 h, demonstrating satisfactory antileishmanicidal activity. Cataneo et al. [28] evaluated the antipromastigote effect of quercetin (up to 192 µM) against L. brasiliensis, in peritoneal cells of macrophages. In promastigote cultures of L. major, quercetin and its analogue quercetin-pentaacetate showed a concentration-dependent effect (IC_50_ = 2.5 ± 0.92 and 2.85 ± 0.99 µM, respectively [24].

Some authors have shown that quercetin induces death in L amazonensis promastigotes through mitochondrial membrane dysfunction resulting from the production of reactive oxygen species [27,28] and other targets, such as arginase [7], ribonucleotide reductase [8,29] and topoisomerase II [30]. The results obtained in this study indicate that other tests should be considered in perspective for the tested compounds (Q and Q5), considering in silico tests for the investigation of targets that may be involved in the observed death mechanisms.

### 2.4. Anti-Inflammatory and Cytotoxicity Activities

Then, we focused on evaluating the biological activity of the novel derivative. First, non-toxic concentrations of Q and Q5 were determined in peritoneal macrophages obtained from BALB/c mice. The CC_50_ values did not demonstrate cytotoxicity at concentrations equal to or less than 80 µM (Figure 2A,D). Dexamethasone was not cytotoxic at the concentration tested (20 µM). Based on this, subsequent tests were carried out at concentrations not exceeding 80 µM.

The immunomodulatory activity of Q and Q5 was investigated to determine the effects of compounds on proinflammatory mediator secretion. The immunomodulatory effects of the compounds (Q and Q5) were initially evaluated in cultures of peritoneal macrophages through the production of nitric oxide. As expected, macrophage activation with LPS PLUS IFNγ increased the amount of nitrite production (Figure 2B,E). Treatment with quercetin inhibited, in a concentration-dependent manner, the production of nitrite (*p* < 0.05).

The pentaacetyl compound (Q5) did not demonstrate similar activity at 20 µM. Interestingly, the activity was significant at the highest concentrations tested (40 and 80 µM), for both compounds. At the highest concentration tested (80 µM), the effects of the compounds were similar to those observed in cultures of activated macrophages and treated with 20 µM dexamethasone. The compounds were not cytotoxic for peritoneal macrophages at the tested concentrations (20, 40 and 80 µM).

For a better characterization of the anti-inflammatory effect of the compounds (Q and Q5), the inflammatory cytokine TNF was quantified by the Enzyme Linked ImmunoSorbent Assay (ELISA) method. Macrophage stimulation using LPS + INFγ induced a prominent increase in TNF production. Treatment with Q and Q5 significantly reduced the production of TNF (*p* < 0.05) in a concentration-dependent manner (Figure 2C,F).

The reduction of inflammatory factors by quercetin is diversely described, including modulation for Th-2 inflammatory profiles, related to protection in neural diseases [31,32]. The anti-inflammatory action of quercetin is well described in the literature [33,34]. However, there are few studies that demonstrate the anti-inflammatory potential of the Q5 analogue. This study corroborates the findings of Chen et al. [35], who evidenced the role of Q and Q5 in the inhibition of the NO production induced by LPS, in a concentration-dependent manner without deleterious cytotoxic effects for the RAW 264.7 macrophage lineage. Furthermore, this study demonstrated the unprecedented effect of Q5 in inhibiting the pro-inflammatory cytokine TNF.

The data obtained indicate the conservation of the effects on the semi-synthetic molecule (Q5); however, the dosage of other factors, such as the cytokine IL-10, can be a perspective in the evaluation of these profiles. Therefore, Q5 can be considered a potential molecule for future in vitro and in vivo tests aiming at an additional therapeutic alternative with anti-inflammatory activity.

The cytotoxicity of the tested compounds (Q and Q5) was evaluated at 20, 40 and 80 µM using the AlamarBlue colorimetric method in a healthy cell line (MRC-5, human lung fibroblasts) and in two different cancer cell lines, HepG2 (human hepatocellular carcinoma) and HL-60 (human promyelocytic leukemia), as shown in Figure 3. Doxorubicin was the standard drug used as a positive control.

As shown in Figure 3, for HepG2 cells, quercetin (Q) did not show significant cytotoxic activity at the highest concentration investigated (80 µM), and its pentaacetate analogue (Q5) showed reduced activity (IC_50_ = 53.9 µM). For the HL-60 cancer cell line, quercetin was shown to be not very active (IC_50_ = 51.3 µM); however, interestingly, Q5 was significantly more active than quercetin with an IC_50_ of 33.6 µM. The standard drug doxorubicin had IC_50_ values ranging from 0.1 to 0.2 µM for cancer cell lines showing significant cytotoxic effects against the healthy cell line MRC-5 (Table 1), which was not seen for the compounds (Q and Q5).

For the HL-60 cell line, the Q and Q5 presented values of 51.3 and 33.6 µM, respectively, in agreement with Massi et al. [17]. Regarding cytotoxicity in non-tumor cells, Q and Q5 presented IC50 values >80 µM, showing a selective profile against the cancer cell line. Few studies have demonstrated the activity of the quercetin Q5 analogue in cancer cell lines. However, Q5 activity in a HeLa tumor cell lineage was documented by Danihelová et al. [22], which showed that acetylated esters of quercetin were the most effective cytotoxic derivatives. No data were found on the cytotoxic role of Q5 in HepG2 lineage cells. Only one study was found in the literature revealing that the tetra-acetylated analogue of quercetin had a significant effect on the inhibition of HL-60 lineage cells through the activation of caspase-3, promoting apoptosis [36]. 

A colorimetric method MTT assay was used to access the cytotoxicity in C6 cell culture. Quercetin and Q5 induced a decrease in 570 nm MTT absorbance that represents mitochondrial dehydrogenase activity and suggests a decrease in the cell viability in C6 cells. As reveled in Figure 4, quercetin-induced cytotoxicity in C6 cells in concentrations higher than 25 µM for 72 h (Figure 4A), while 0.78 µM of Q5 for 72 h was able to decrease cell viability (Figure 4B). It was observed that 50 µM quercetin (Q) and Q5 decreased the cell viability to 41.3% ± 2.9% and 47.5% ± 1.2%, respectively, when compared with the control group (100.0% ± 6.4%) (Figure 4A,B). No significant changes in cell viability were visualized in C6 cultures treated with DMSO (0.05%), when compared with untreated cells.

Based on the results obtained, we then investigated the morphological effects of compounds Q and Q5 (50 µM) on the C6 cells, at 24, 48 and 72 h of the treatment, using optical microscopy. Quercetin pentaacetate (Q5) at 50 µM induced changes in the morphology of C6 glioma cells within the first 24 h, with a visible reduction in cytoplasmic prolongations when compared to the control group (Figure 5). After 48 h of treatment with Q5, the cells assumed a rounded morphology characterized by retraction of the cell body and shapeless membrane (which was not reported in the scientific literature), unlike quercetin, which, during this time of treatment, presented a morphological aspect similar to fibroblasts [37]. Any other morphological changes were visualized in cells under other treatment conditions.

The hydroxyl substitution by the acetyl groups may promote improved cellular absorption of the quercetin analogues favoring various biological tests, particularly in cancer cells [38,39]. Bispo da Silva et al. [40] demonstrated, by treating C6 cells with the flavonoids rutin and quercetin, a significant reduction in the proportion of adherent C6 cells, with a thinner and bipolar morphological phenotype, compared to control cultures. Treatment of C6 cells with flavonoids inhibited the migratory properties of viable C6 cells after 24 h of treatment. In control conditions, microglia presented a rounder phenotype; after rutin treatment, more than 50% of cells acquired a branched, multipolar phenotype, and the others acquired the amoeboid phenotype, both indicating activation. 

Studies indicate that quercetin interferes in the regulation of cell signaling transduction pathways associated with cell death by apoptosis and in the stages of cell cycle progression [41,42]. Santos et al. [14] indicated a potential reduction in the growth of GL-15 human glioblastoma cells. Bi et al. [43] visualized an induction of autophagy of U87 and U251 human glioblastoma cells in a dose-dependent manner.

The results obtained corroborate with Danihelova et al. [22], in which the acetyl groups inserted in the quercetin molecule promoted an improvement in anticancer activity. Furthermore, they indicated a greater tropism for cancer cells, particularly for cells of neural lineages. This study is unprecedented in relation to the Q5 analogue in the treatment of C6 glioma cells. Dell’Albani et al. [44] showed a better action of acyl derivatives of quercetin in cultures of human glioma strains U373-MG and murine glioma 9L, indicating pathways of death by apoptosis. The modification of cell morphology to a rounded profile can refer to death mechanisms, such as apoptosis, widely attributed to quercetin [45,46,47]. In future perspectives, tests with healthy neural cells may help to elucidate this hypothesis.

## 3. Materials and Methods

### 3.1. Reagents and Materials

All reagents (quercetin, pyridine, acetic anhydride, tricoloroisocyanuric acid, dichloromethane, 2,4-dinitro-phenylhydrazine, hydroxylamine hydrochloride, sodium acetate, petroleum ether, sulfuric acid, potassium persulfate and nitrate) were analytical grade and commercially available (Quimex^®^, Merck, Brazil and Sigma Aldrich^®^, St. Louis, MO, USA). For the preparation of all standard solutions and samples, ultrapure water (with resistivity 18 MΩ cm^−1^) obtained from the Milli-Q Pluswater purification system (Millipore, Molsheim, France) was used. All laboratory glassware was washed in a 10% (*v*/*v*) HNO_3_ solution for 24 h, rinsed with high-purity water and dried at ambient temperature.

### 3.2. Synthesis and Characterization

The structural analog was obtained after quercetin molecular modification from the more accessible and reproducible synthetic route (penta acetylation). We applied the green chemistry principles with minimization of substance use. The pentaacetyl analog (Q5) was obtained after the molecular modification of quercetin, from a synthetic route, which uses acetic anhydride as an acylating agent and pyridine as a catalyst.

The mixture containing quercetin (300 mg, 1 eq.), acetic anhydride (0.80 mL, 20 eq.) and pyridine (7.5 mL) was kept under magnetic stirring at room temperature. After 24 h of stirring, 250 mL of dichloromethane was added to the reaction medium. Then, the reaction was washed with 10% HCl (3 × 100 mL), diluted NaOH (3 × 50 mL) and water (3 × 100 mL); dried over anhydrous sodium sulfate; filtered; and evaporated, in the rotary evaporator (Fisatom, Minas Gerais, Brazil) [21]. The amount of Q5 and the yield obtained were 280 mg and 54%, respectively.

The structural modification reaction of the quercetin molecule was monitored by thin layer chromatography (TLC) using a mixture of hexane and ethyl acetate (1:1) as solvents. For the physicochemical characterization, the melting point was used. Fourier Transform Infrared (FTIR) tests by the Attenuated Total Reflection (ATR) method were performed by a FTIR Spectrum 100S model (Perkin Elmer, Waltham, MA, USA), with the acquisition of the scanning spectra in the mid-infrared (4000 to 600 cm^−1^).

The sample (mass ~5.0 mg) was placed on the ATR crystal and subjected to a pressure of approximately 20 N with the aid of a manual mechanical press. FTIR spectra were traced by OriginPro8 software, OriginLAB^®^ (www.originlab.com, accessed in 10 September 2021). Nuclear Magnetic Resonance (NMR) spectra were recorded in a Bruker Avance III 500 MHz (Uster, Switzerland) spectrometer-500 MHz for ^1^H NMR and 125 MHz for ^13^C NMR, using DMSO as a solvent. Chemical shifts were expressed in the ppm scale (µg/mL), and chloroform (CHCl_3_) was used as the internal reference. 

NMR spectra were processed by TopSpin^®^ 4.0 software (Bruker Biospin, Coventry, UK) and compared with literature data [38]. The NMR data were: ^1^H NMR (500 MHz, CDCl_3_) 2.34 (6 H, s, –OCOCH_3_); 2.35 (3 H, s, –OCOCH_3_); 2.35 (3 H, s, –OCOCH_3_); 2.44 (3 H, s, –OCOCH_3_); 6.88 (1 H, d, *J* = 2.5 Hz); 7.34 (1 H, d, *J* = 2.0 Hz); 7.36 (1 H, d, *J* = 8.5); 7.70 (1 H d, *J* = 2.0 Hz); 7.73 (1H, dd, *J*_1_ = 8.5 Hz, *J*_2_ = 2.0 Hz); ^13^C NMR (125 MHz, CDCl_3_) δ 170.04; 169.24; 167.86; 167.79; 156.87; 154.28; 150.43; 144.40; 142.22; 127.78; 126.42; 124.01; 123.93; 123.85; 113.89; 108.96; 21.16; 21.02; 20.65; and 20.49.

### 3.3. Antioxidant ABTS^•+^ Radical Activity

The ABTS^•+^ sequestration activity was adapted from the method described by Dorman and Hiltunen [48]. Potassium persulfate and ABTS (Sigma Aldrich^®^, St. Louis, MO) were dissolved in distilled water to form a final concentration of 2.45 mM and 7 mM, respectively. The ABTS^•+^ solution was produced by adding both solutions at a rate of 1:1, and then the solution was incubated at room temperature for 16 h in the dark. 

The resulting solution, intensely colored, was adjusted with ethanol, in the spectrophotometer, to an absorbance of 0.7 ± 0.05 nm at 734 nm before use. We used 30 µL of the samples, or Trolox (Sigma Aldrich^®^, St. Louis, MO), as a reference standard at different concentrations (5, 10, 25, 50, 75 and 100 µM), were added to 3 mL of the ABTS^•+^ solution and awaited to reacts for 6 min. The absorbance was measured at 734 nm against a blank (ethanol). The ABTS^•+^ scavenging capacity was calculated as: ABTS^•+^ scavenging effect (%) = (1 − A0/A1) × 100;
where A0 is the absorbance of the control, and A1 is the absorbance of the sample or standard. All determinations were performed in triplicate. IC_50_ values were calculated and expressed as the mean ± SD in µM.

### 3.4. Antileishmania Activity

*L. amazonensis* (MHOM/BR88/BA-125 Leila strain) and *L. braziliensis* (MHOM/BR88/BA-3456) promastigotes (1 *×* 10^6^ per well) were cultured in a 96-well plate in Schneider medium (Sigma-Aldrich^®^, St. Louis, MO, USA) supplemented with 10% fetal bovine serum (FBS; GIBCO) and 50 μg mL^−^^1^ gentamicin (Life, Carlsbad, CA, USA) and subjected to treatment with different concentrations (100 μM; six dilutions 1:2) of Q5. The parasites were incubated for 72 h at 26 °C. Then, 20 µL/well of AlamarBlue (Invitrogen, Carlsbad, CA, USA) was added over 2 h. The reading was carried out in a spectrophotometer using the wavelengths of 570 and 600 nm. The calculation of axenic culture inhibition was determined based on the untreated control [49]. 

### 3.5. Anti-Inflammatory and Cytotoxicity Activities

#### 3.5.1. Drugs

Dexamethasone (Sigma-Aldrich^®^, St. Louis, MO, USA), a synthetic glucocorticoid, was used as positive control in immunomodulatory assays. Doxorubicin (doxorubicin hydrochloride, Laboratory IMA S.A.I.C., Buenos Aires, Argentina) was used as a reference anticancer drug. All compounds were dissolved in dimethyl sulfoxide (DMSO; PanReac, Barcelona, Spain) and diluted in cell culture medium for use in the assays. The final concentration of DMSO was less than 1% in all experiments.

#### 3.5.2. Animals

BALB/c mice, aged 4–10 weeks, were provided by the vivarium of Gonçalo Moniz Institute/FIOCRUZ-BA (IGM, Salvador, Bahia, Brazil) where they were kept in cages containing a maximum of five mice. All cages were kept in an acclimatized room at 21 °C ± 1 °C, on a 12-h light/dark cycle, with water and food ad libitum throughout the experimental period. The experiments with animals were approved by the Ethics and Animal Use Committee of Gonçalo Moniz/FIOCRUZ-BA Institute (IGM-018/15).

#### 3.5.3. Cells 

HL-60 (human acute promyelocytic leukemia) and HepG2 (human hepatocellular carcinoma) were obtained from the American Type Culture Collection—ATCC (Rocville, ML, USA), were used. To assess the selectivity of quercetin (Q) and the Q5 derivative on the proliferation of non-cancer cells, the MRC-5 lineage (human lung fibroblast), also obtained from ATCC, was used. Cell lines were grown in cell culture bottles (75 cm^3^, 250 mL volume) using RPMI 1640 culture medium (Gibco™) supplemented with 10% fetal bovine serum (Gibco™) and 50 µg/mL gentamicin (Gibco™). The cells were kept in incubators with an atmosphere of 5% CO_2_ at 37 °C and monitored daily. All cell lines were tested for mycoplasma using a Hoechst staining mycoplasma detection kit (Sigma-Aldrich, St. Louis, MO, USA).

C6 cell line was derived from rat glial tumors induced by *n*-nitrosomethylurea [50]. This glioma cells were cultured as described by de Oliveira et al. [51]. The cells were incubated at 37 °C in a humidified incubator at 5% CO_2_. C6 cells were grown on cell culture dishes (100-mm Ø, TPP) in DMEM medium supplemented with 100 UI/mL penicillin G, 100 µg/mL streptomycin, 7 mM glucose, 2 mM L-glutamine, 1 mM pyruvate and 10% fetal calf serum. The culture medium was changed every 2 days. Twenty-four hours prior to treatments, C6 cells were seeded in Petri dishes of 35 mm in diameter or 96-well plates at a density of 3.5 × 10^4^ cells/well.

All cell lines were tested for mycoplasma using the Mycoplasma Stain Kit (Sigma-Aldrich) to validate the use of cells free from contamination. The cell lines information is contained in the database “Cell Bank of Rio de Janeiro (BCRJ)”: HepG2 cell (Code: 0291), HL-60 cell (Code: 0104) and C6 cell (Code: 0057).

#### 3.5.4. Cytotoxic Activity Assay

To assess the cytotoxicity of the tested compounds (Q and Q5) on the HL-60 (human acute promyelocytic leukemia) and HepG2 cells, the colorimetric method of Alamar Blue (Invitrogen, Carlsbad, CA, USA) was used. Alamar Blue (resazurin) is an indicator that produces a colorimetric change and a fluorescent signal in response to metabolic activity. Resazurin is reduced to resorufin by metabolically active cells. The oxidized form is blue (non-fluorescent/non-viable cell), and the reduced form is pink (fluorescent/viable cell). The reduction of resazurin to resorufin reflects the cell viability [52]. 

In the assay, cells were distributed in 96-well plates at a predefined density of 0.3 × 10^6^ cells/mL for cells of the HL-60 lineage and 0.7 × 10^5^ cells/mL for cells of the HepG2 lineage. The compounds were added in a series of eight concentrations (80 to 0.62 µM), with the exception of doxorubicin, which was used in concentrations ranging from 0.003 to 5 µM. The negative control received the same amount of DMSO (0.025%). The plates were incubated for 72 h in an oven at 37 °C and 5% CO_2_. After this period, 20 μL/well of Alamar Blue was added, and the plates were incubated for another 4 h. The plates were read using a spectrophotometer (Spectramax 190, Molecular Devices, Sunnyvale, CA, USA), at wavelengths of 570 and 600 nm.

The C6 cell viability was determined using the colorimetric method described by Hansen et al. [53]. The MTT (3–4,5-dimethylthiazol-2-yl, 2,5-diphenyltetrazolium bromide) was dissolved at a concentration of 5 mg/mL in sterile phosphate buffered saline (PBS) at room temperature, and the solution was further sterilized by passing through a 0.2-mm filter and stored at 4 °C in the dark. In the assay, cells were distributed in 96-well plates at a predefined density of 3.5 × 10^4^ cells/mL for the C6 cell lineage. The compounds were added in a series of eight concentrations (80 to 0.39 µM). 

MTT was added in each well at a final concentration of 2 mg/mL, and the cells were incubated for 2 h. After that, cells were lysed with 20% (*w*/*v*) sodium dodecyl sulfate (SDS) and 50% (*v*/*v*) dimethylformamide (DMF) solution (pH 4.7), in an overnight incubation at room temperature [54,55]. The absorbance was measured with a spectrophotometer microplate reader (570 nm), Thermo Scientific^®^ Flash Varioskan (Version 3001, Thermo Fisher Scientific, Vantaa, Finland). Eight replicates were used for each concentration. Cell viability was expressed as the percentage of absorbance at 570 nm, and the control was adopted as 100%. After treatments, cell morphology was evaluated by light microscopy using an optic microscope (Eclipse TS100 inverted microscope, Nikon Instruments, Tokyo, Japan) and photographed using a digital camera (Coolpix S4300, Nikon Instruments, Tokyo, Japan).

#### 3.5.5. Macrophage Culture

Peritoneal exudate macrophages (2 × 10^5^ cells/well) were incubated in 96-well plates in DMEM medium supplemented with 10% PBS and 50 µg/mL gentamicin, in triplicate, stimulated or not with LPS (500 ng/mL) and IFN-γ (5 ng/mL) and treated or not with different concentrations of compounds (20, 40 and 80 µM). The cells were kept in an incubator at 37 °C and 5% CO_2_ for 4 and 24 h. After this period, the culture supernatants were collected for cytokines and nitric oxide dosage. In some assays, to assess the cytotoxicity, the supernatant was replaced with medium plus 10% Alamar Blue (Invitrogen, Carlsbad, CA, USA), and the plates were incubated for an additional 4 h. The spectrophotometer reading was performed at 570 and 600 nm.

#### 3.5.6. TNF Dosage

TNF measurement was performed from cell culture supernatants, using the sandwich ELISA technique, using Development System kitsDuoset ELISA (R&D Systems, Minneapolis, MI, USA), according to the manufacturer’s recommendations. ELISA plates (NUNC—IMMUNO PLATE Maxisorp Surface) were sensitized with 50 µL/well of the capture antibody, at a concentration of 2 µg/mL, diluted in PBS 1 x and incubated for 16 h at 4 °C. The plates were washed three times with 1 × PBS/0.05% tween 20 and blocked with 100 µL/well of 1 × PBS and 0.05% tween 20 and 0.1% bovine albumin for 2 h.

Then, 50 µL/well of samples, blank and standard curve of recombinants diluted in Tris-saline buffer (20 mM trix in the base and 150 mM NaCl) containing 0.1% bovine albumin and 0.05 were added % tween 20 for 2 h at room temperature. The standard was serially diluted (1:2), from the initial concentration of 2000 pg/mL, with 11 duplicate dilutions. The plate was washed three times with PBS/0.05% tween and incubated with 50 µL of the detection antibody (biotinylated) at a concentration of 400 ng/mL for a period of 2 h.

The plate was washed three times with PBS/0.05% tween and incubated for 20 min with avidin-peroxidase diluted 1:200. The development was carried out by adding TMB substrate (Thermo Fisher) and interrupted with 0.05 M phosphoric acid. The reading of the reaction was determined using a spectrophotometer (Spectramax) (Molecular Devices, San Jose, CA, USA), with a 450 nm filter. Analyses were performed using the Softmax 4.3.1 Software (Molecular Devices, San Jose, CA, USA).

#### 3.5.7. Nitric Oxide Dosage

The production of nitrite in macrophage supernatants was estimated through the quantification of its oxidative product, nitrite, by the Griess method [50]. The absorbance was determined in a spectrophotometer (Spectramax) (Molecular Devices, San Jose, CA, USA), with a 570 nm filter. Analyses were performed using Softmax Software 4.3.1 (Molecular Devices, San Jose, CA, USA), and the results were expressed in µM of nitrite, based on a standard curve of sodium nitrite with an initial concentration of 400 µM.

### 3.6. Statistical Analysis

The one-way ANOVA test followed by the Bonferroni multiple comparison post-test was used to determine the statistical significance of the comparisons between groups in the studies. The results were considered statistically significant when *p* < 0.05. All analyses were performed using the GraphPad Prism version 5.01 program (GraphPad Software, San Diego, CA, USA).

## 4. Conclusions

The conditions and reactions synthesis media of quercetin analogues showed efficiency in obtaining one compound duly characterized, according to the data described in the literature. The comparative analysis of the antioxidant activity of Q and Q5 showed that quercetin (Q) was more active in scavenging the ABTS^•+^ radical than Q5 (29% and 18%, respectively). The reduced antioxidant potential did not demonstrate interference in immunomodulatory and antitumoral activities. The chemical modifications proposed in this study enhanced the antiproliferative effect and maintained the anti-inflammatory activity but not cytotoxicity activity in healthy tested cells.

The acetylated derivative (Q5) present improved the cytotoxicity in cancer hepatocellular cells (HepG2), promyelocytic leukemia (LH-60) cells and, particularly, in glioma (C6) cells. The rat glioma C6 cells traded showed a morphological unprecedent pattern of cell death. Q5 at 50 µM for 24 h induced changes in C6 glioma cell morphology characterized by a round body shape (which was not reported in the scientific literature), unlike quercetin, which presented a fibroblast-like morphology.

Further investigations need to be done to better understand the effects of acetylated derivatives of quercetin, particularly in neuronal cells. This study allowed, for the first time, the application of structurally modified quercetin in neural cells for potential neuroprotective effects.

## Figures and Tables

**Figure 1 molecules-26-06923-f001:**
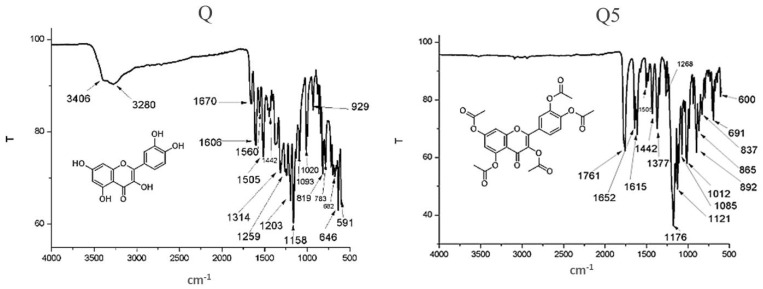
FTIR spectrum of the compounds Quercetin (Q) and Quercetin pentaacetate (Q5), demonstrating the disappearance of representative hydroxyl bands next to 3280 cm^−1^ and the appearance of bands close to 1761 cm^−1^ assigned to acetyl groups.

**Figure 2 molecules-26-06923-f002:**
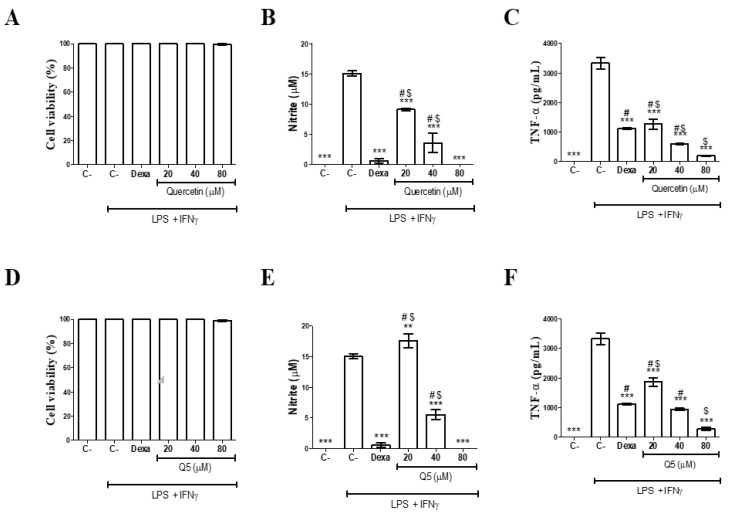
Effects of quercetin (Q) and quercetin-penta acetate analogue (Q5) on macrophages (in vitro). Peritoneal exudate macrophages stimulated or not with LPS + INFγ were cultured in the presence or absence of compounds (20, 40 or 80 µM) or dexamethasone (20 µM). Cell viability was determined by the Alamar Blue method (**A**,**D**). Cell supernatant was collected after 24 h for nitrite quantification (**B**,**E**) or 4 h for TNF measurement (**C**,**F**). C- Group of untreated and unstimulated cells. C- Group of cells stimulated with LPS + INFγ. Values are represented by the mean ± standard deviation of the mean of nine determinations obtained from three independent experiments. *** *p* < 0.001 compared to stimulated and untreated cells, ** *p* < 0.01 compared to stimulated and untreated cells, # *p* < 0.05 compared to unstimulated and untreated cells and $ *p* < 0.05 compared to cells treated with dexamethasone.

**Figure 3 molecules-26-06923-f003:**
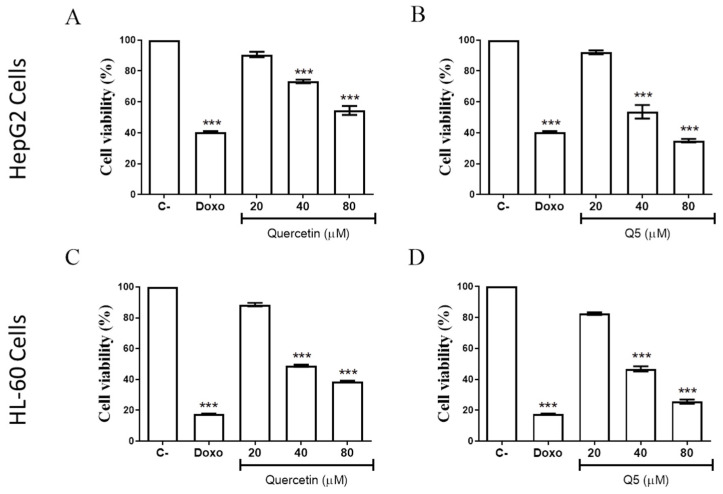
Effects of quercetin (Q) and the quercetin-pentaacetate analogue (Q5) on the viability of HepG2 (**A**,**B**) and HL-60 (**C**,**D**) cells (20, 40 and 80 µM) or doxorubicin (5 µM) as determined by AlamarBlue after 72 h of treatment. Values are represented as the mean ± SD of three independent experiments performed in triplicate. *** *p* < 0.001 compared to untreated cells.

**Figure 4 molecules-26-06923-f004:**
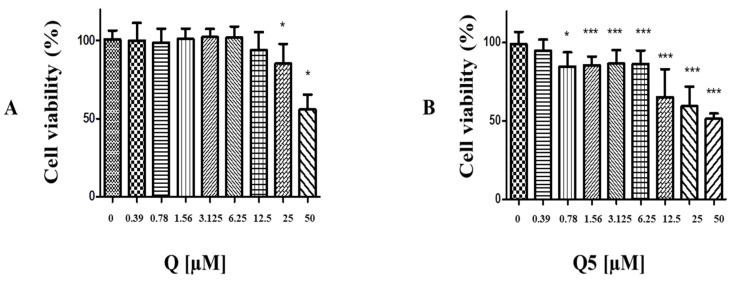
Analysis of cytotoxic activity by MTT test in C6 cells exposed to Q and Q5 compound at concentrations of 50 µM and 7 1:2 dilutions. (**A**) C6 after 72 h of exposure to quercetin. (**B**) C6 after 72 h of exposure to quercetin pentaacetate (Q5). Cells under control conditions were treated with 0.05% DMSO—a vehicle for drug dilution. The results expressed as a percentage in relation to the control, taken as 100% (* *p* < 0.05; *** *p* < 0.001). The different patterns of the column chart represent different concentrations of quercetin (Q) in (**A**), and quercetin pentaacetate (Q5) in (**B**).

**Figure 5 molecules-26-06923-f005:**
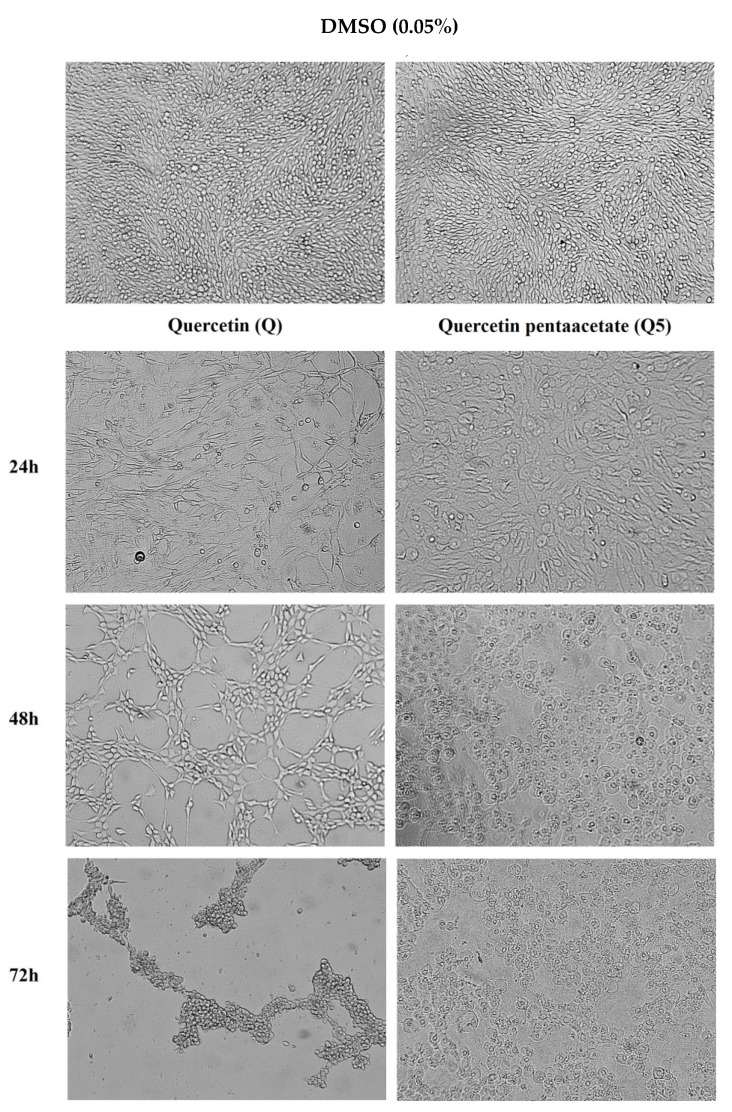
Morphological changes of C6 cells under control conditions (0.05% DMSO), after 24, 48 and 72 h of exposure to 50 µM of quercetin (Q) or quercetin-pentaacetate (Q50) compounds.

**Table 1 molecules-26-06923-t001:** Cytotoxicity of the compounds Quercetin (Q) and Quercetin penta-acetate (Q5) against the healthy cell line MRC-5 (human lung fibroblasts) and different cancer cell lines: HepG2 (human hepatocellular carcinoma), HL-60 (human promyelocytic leukemia) and C6 (rat glioma).

	Tested Compounds (IC_50_ µM)		
	Quercetin (Q)	Q5	Doxorrubicin
Cancer cells			
IC_50_ (µM) ^a^ HL-60	51.3 (±0.4)	33.6 (±2.6)	0.2 (±0.0)
IC_50_ (µM) ^b^ HepG2	>80	53.9 (±11.3)	0.1 (±0.0)
Non-cancer cells			
CC_50_ MRC-5	>80	>80	0.9 (±0.0)

^a^ Determined in HL-60 incubated with compounds for 72 h; ^b^ determined in HepG2 incubated with compounds for 72 h. Values represented as the mean ± SD and were calculated using three independent experiments. CC_50_ = 50% cytotoxic concentration. IC_50_ = 50% inhibitory concentration.

## Data Availability

Not applicable.

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
