# Peer review of "Structural Design, Synthesis and Antioxidant, Antileishmania, Anti-Inflammatory and Anticancer Activities of a Novel Quercetin Acetylated Derivative"

_molecules, 2021, doi:10.3390/molecules26226923_

Round 1
Reviewer 1 Report
This paper describe synthesis of quercetin pentaacetate analogue, in order to investigate its biological potentials in particular antioxidant, antileishmania, anti-inflammatory and cytotoxicity activities.
All the significant studies for characterization (NMR, IR, melting point) were made. While study described is sufficient, number derivative studied is little and would be interesting compare at least another acetate to quercetin and not only one.
Pls add peaks values on figure NMR (S1, S2 , S4, S5 , S7).
Also if authors performed NMR and IR (with evidence of absence of OH groups) , characterization of pentaacetate would request Mass spectra to confirm MW of derivative. Pls add this mass spectra of compound and % purity of it.
Antioxidant ABTS•+ radical activity , Antileishmania activity, Anti-inflammatory and cytotoxicity activities were performed to demonstrate their conclusion.  Optical Microscopical was used to investigate the morphological effects of compounds Quercetin and Quercetin pentaacetate on the C6 cells, in 72 hours of the treatment. These photo should be improved because not much clear.
I suggest to update reference list
Also explain better concept relative to morphology change of glioma because not much clear ( line 360-369).
Authors are advised to check the grammatical errors throughout the manuscript .
Hence, this reviewer indicate accept this MS for publication after major revision in Molecules.   
Author Response
Response to Reviewers’ Comments
Structural design, synthesis and biological activities (antioxidant, antileishmania, anti-inflammatory and anticancer) of novel quercetin acetylated derivative
Saul Vislei Simões da Silva, Orlando Maia Barboza, Jéssica Teles Souza, Érica Novaes Soares, Cleonice Creusa dos Santos, Luciano Vasconcellos Pacheco, Ivanilson Pimenta Santos, Tatiana Barbosa dos Santos Magalhães, Milena Botelho Pereira Soares, Elisalva Teixeira Guimarães, Cássio Santana Meira, Silvia Lima Costa, Victor Diógenes Amaral da Silva, Lourenço Luís Botelho de Santana and Aníbal de Freitas Santos Júnior
We would like to thank Reviewers for valuable comments regarding the submitted manuscript. All changes made in paper are in red colour.
Reviewer 1.
All comments on editorial, grammar and content errors have been improved.
* Reviewer 1: Comments and Suggestions for Authors
This paper describe synthesis of quercetin pentaacetate analogue, in order to investigate its biological potentials in particular antioxidant, antileishmania, anti-inflammatory and cytotoxicity activities.
- All the significant studies for characterization (NMR, IR, melting point) were made. While study described is sufficient, number derivative studied is little and would be interesting compare at least another acetate to quercetin and not only one. Response: Thanks for the comment! The initial idea was this! In principle, we obtained tetra and pentaacetyl derivatives. However, we chose to continue the studies with pentaacetylated due to the reaction yield and better performance obtained in preliminary experiments. For future studies, we will expand the synthetic routes to obtain other derivatives.
- Pls add peaks values on figure NMR (S1, S2 , S4, S5 , S7). Response: Revised!
- Also if authors performed NMR and IR (with evidence of absence of OH groups) , characterization of pentaacetate would request Mass spectra to confirm MW of derivative. Pls add this mass spectra of compound and % purity of it. Response: Thanks for the comment! Analysis by mass spectrometry would be one more indication that the target molecule was obtained, but the synthesis and purification of the compound in question was carried out through a complete reproduction of the process described in the literature, with all physical data, melting point, IR, NMR compatible with those reported in the base work, which makes the identification of the substance unambiguous. In addition, considering the current pandemic moment of the New Coronavirus in Brazil and the restrictions on access to laboratories in our city, it is impossible to obtain an additional quantity of the molecule and perform the requested analysis.
- Antioxidant ABTS•+ radical activity, Antileishmania activity, Anti-inflammatory and cytotoxicity activities were performed to demonstrate their conclusion.  Optical Microscopical was used to investigate the morphological effects of compounds Quercetin and Quercetin pentaacetate on the C6 cells, in 72 hours of the treatment. These photo should be improved because not much clear. Response: Revised!
I suggest to update reference list. Response: Revised! More references were added and discussed!
- Also explain better concept relative to morphology change of glioma because not much clear (line 360-369). Response: Revised!
- Authors are advised to check the grammatical errors throughout the manuscript. Response: All comments on grammar and content errors have been improved.
Hence, this reviewer indicate accept this MS for publication after major revision in Molecules.   
Reviewer 2 Report
Corrections /Additions needed.
line 25 limit its biopharmacological use.
line 29 1H and 13C NMR spectra.
line 36 by acetyl groups.
line 44 anti no hyphen
line 78 peripheral mononuclear cells (PBMC) acronym does not fit!
line 104 and by the appearance
line 115 RMN What is this full name should be given
lines 123 to 127 NMR data should be given in the Experimental Section
line 128 The spectra are compatible
lines134 to 138 The whole statement it does not makes sense. Re-write it in better English
line 200 pentaacetyl
line 398 TLC (give mobile phase ie solvents)
lines 387 to 396 Synthesis ... give amount of product obtained and % yield.
line 549 showed efficiency in obtaining one...
Author Response
Response to Reviewers’ Comments
Structural design, synthesis and biological activities (antioxidant, antileishmania, anti-inflammatory and anticancer) of novel quercetin acetylated derivative
Saul Vislei Simões da Silva, Orlando Maia Barboza, Jéssica Teles Souza, Érica Novaes Soares, Cleonice Creusa dos Santos, Luciano Vasconcellos Pacheco, Ivanilson Pimenta Santos, Tatiana Barbosa dos Santos Magalhães, Milena Botelho Pereira Soares, Elisalva Teixeira Guimarães, Cássio Santana Meira, Silvia Lima Costa, Victor Diógenes Amaral da Silva, Lourenço Luís Botelho de Santana and Aníbal de Freitas Santos Júnior
We would like to thank Reviewers for valuable comments regarding the submitted manuscript. All changes made in paper are in red colour.
Reviewer 2.
* Reviewer 2: Comments and Suggestions for Authors
Corrections /Additions needed. Response: All comments on editorial, grammar and content errors have been improved.
- line 25 limit its biopharmacological use. Response: Revised!
- line 29 1H and 13C NMR spectra. Response: Revised!
- line 36 by acetyl groups. Response: Revised!
- line 44 anti no hyphen. Response: Revised!
- line 78 peripheral mononuclear cells (PBMC) acronym does not fit! Response: Revised! Changed to peripheral blood mononuclear cells (PBMC).
- line 104 and by the appearance. Response: Revised!
- line 115 RMN What is this full name should be given. Response: Revised!
- lines 123 to 127 NMR data should be given in the Experimental Section. Response: Revised! The data has been moved to the "Materials and Methods" section.
- line 128 The spectra are compatible. Response: Revised!
- lines 134 to 138 The whole statement it does not makes sense. Re-write it in better English.. Response: Thanks! The text was reformulated and adapted in another section of the manuscript.
- line 200 pentaacetyl. Response: Revised!
- line 398 TLC (give mobile phase ie solvents). Response: Revised!
- lines 387 to 396 Synthesis ... give amount of product obtained and % yield. Response: Revised! The information has been entered in section “2. Results and Discussion - 2.1 Synthesis and characterization”.
- line 549 showed efficiency in obtaining one... Response: Revised!
Round 2
Reviewer 1 Report
This paper was improved after suggestions.
Hence this reviewer indicate accept this MS for publication in Molecules.
Author Response
Response to Reviewers’ Comments
Structural design, synthesis and biological activities (antioxidant, antileishmania, anti-inflammatory and anticancer) of novel quercetin acetylated derivative
Saul Vislei Simões da Silva, Orlando Maia Barboza, Jéssica Teles Souza, Érica Novaes Soares, Cleonice Creusa dos Santos, Luciano Vasconcellos Pacheco, Ivanilson Pimenta Santos, Tatiana Barbosa dos Santos Magalhães, Milena Botelho Pereira Soares, Elisalva Teixeira Guimarães, Cássio Santana Meira, Silvia Lima Costa, Victor Diógenes Amaral da Silva, Lourenço Luís Botelho de Santana and Aníbal de Freitas Santos Júnior
We would like to thank Reviewers for valuable comments regarding the submitted manuscript.
Reviewer 1: No corrections were requested.
Reviewer 2 Report
line 195 80 micro (symbol) M needs space
line 319 48 e 72 hours change to 40 and 72 hours
lines 399 to 403 chemical shifts should be reported with a . ( not ,) indicating decimal place. Also tow decimal places are usually reported for 1H nMR spectra.
Author Response
Response to Reviewers’ Comments
Structural design, synthesis and biological activities (antioxidant, antileishmania, anti-inflammatory and anticancer) of novel quercetin acetylated derivative
Saul Vislei Simões da Silva, Orlando Maia Barboza, Jéssica Teles Souza, Érica Novaes Soares, Cleonice Creusa dos Santos, Luciano Vasconcellos Pacheco, Ivanilson Pimenta Santos, Tatiana Barbosa dos Santos Magalhães, Milena Botelho Pereira Soares, Elisalva Teixeira Guimarães, Cássio Santana Meira, Silvia Lima Costa, Victor Diógenes Amaral da Silva, Lourenço Luís Botelho de Santana and Aníbal de Freitas Santos Júnior
We would like to thank Reviewers for valuable comments regarding the submitted manuscript. All changes made in paper are in red colour.
* Reviewer 2: Comments and Suggestions for Authors
- line 195 80 micro (symbol) M needs space. Response: Revised!
- line 319 48 e 72 hours change to 40 and 72 hours. Response: Revised!
- lines 399 to 403 chemical shifts should be reported with a . ( not ,) indicating decimal place. Also tow decimal places are usually reported for 1H nMR spectra. Response: Revised in the manuscript and Supplementary information!